# The cytokines HGF and CXCL13 predict the severity and the mortality in COVID-19 patients

Matthieu Perreau[1,11], Madeleine Suffiotti[1,11], Pedro Marques-Vidal [2], Aurelie Wiedemann [3,4], Yves Levy [3,4], Cédric Laouénan[5,6], Jade Ghosn[7], Craig Fenwick[1], Denis Comte[1], Thierry Roger [8], Jean Regina [8], Peter Vollenweider[2], Gerard Waeber [2], Mauro Oddo[9], Thierry Calandra[8] & Giuseppe Pantaleo [1,3,10 ✉]

The objective of the present study was to identify biological signatures of severe coronavirus disease 2019 (COVID-19) predictive of admission in the intensive care unit (ICU). Over 170 immunological markers were investigated in a 'discovery' cohort (n = 98 patients) of the Lausanne University Hospital (LUH-1). Here we report that 13 out of 49 cytokines were significantly associated with ICU admission in the three cohorts (P < 0.05 to P < 0.001), while cellular immunological markers lacked power in discriminating between ICU and non-ICU patients. The cytokine results were confirmed in two 'validation' cohorts, i.e. the French COVID-19 Study (FCS; n = 62) and a second LUH-2 cohort (n = 47). The combination of hepatocyte growth factor (HGF) and C-X-C motif chemokine ligand 13 (CXCL13) was the best predictor of ICU admission (positive and negative predictive values ranging from 81.8% to 93.1% and 85.2% to 94.4% in the 3 cohorts) and occurrence of death during patient follow-up (8.8 fold higher likelihood of death when both cytokines were increased). Of note, HGF is a pleiotropic cytokine with anti-inflammatory properties playing a fundamental role in lung tissue repair, and CXCL13, a pro-inflammatory chemokine associated with pulmonary fibrosis and regulating the maturation of B cell response. Up-regulation of HGF reflects the most powerful counter-regulatory mechanism of the host immune response to antagonize the pro-inflammatory cytokines including CXCL13 and to prevent lung fibrosis in COVID-19 patients.

[1] Service of Immunology and Allergy, Lausanne University Hospital, University of Lausanne, Lausanne, Switzerland. [2] Service of Internal Medicine, Department of Medicine, Lausanne University Hospital, University of Lausanne, Lausanne, Switzerland. [3] Vaccine Research Institute, Université Paris-Est, Faculté de Médecine, INSERM U955, Créteil, France. [4] Assistance Publique-Hôpitaux de Paris, Groupe Henri-Mondor Albert-Chenevier, Service d'Immunologie Clinique, Créteil, France. [5] AP-HP, Hôpital Bichat, Département Épidémiologie Biostatistiques et Recherche Clinique, INSERM, Centre d'Investigation clinique-Epidémiologie Clinique 1425, Paris, France. [6] Université de Paris, INSERM, IAME UMR 1137, Paris, France. [7] AP-HP, Hôpital Bichat, Service de Maladies Infectieuses et Tropicales, Paris, France. [8] Service of Infectious Diseases, Lausanne University Hospital, University of Lausanne, Lausanne, Switzerland. [9] Service of Intensive Care, Lausanne University Hospital, University of Lausanne, Lausanne, Switzerland. [10] Swiss Vaccine Research Institute, Lausanne University Hospital, University of Lausanne, Lausanne, Switzerland. [11]These authors contributed equally: Matthieu Perreau, Madeleine Suffiotti. ✉email: Giuseppe.Pantaleo@chuv.ch

Severe acute respiratory syndrome coronavirus 2 (SARS-CoV-2), the cause of coronavirus disease 19 (COVID-19) induces a broad range of clinical manifestations including asymptomatic infection, mild disease, and a life-threatening severe clinical syndrome characterized by respiratory failure, shock, and multi-organ dysfunction requiring admission in the intensive care unit (ICU). The severe COVID-19 is associated with a mortality of 5–10%[1–3].

Several studies have hypothesized that the severity of COVID-19 results from an excessive inflammatory immune response that may cause a life-threatening multi-organ systemic clinical syndrome[4–6]. Similar to SARS-CoV, the inflammatory innate response is mainly due to a massive cytokine and chemokine release syndrome[7,8]. Patients with COVID-19 have elevated serum levels of cytokines (interleukin-1 (IL-1), IL-6, IL-2, IL-7, IL-10, IL-12, and IFN-γ,), chemokines (CCL2, CCL3, CXCL8, CXCL9, and CXCL10, CXCL11), and growth factors (G-CSF and hepatocyte growth factor (HGF))[9,10]. Recently, Marie Del Valle et al. showed in a retrospective analysis that high serum levels of IL-6, IL-8, and TNF-α at the time of hospitalization were strong and independent predictors of patient survival[11]. Additional studies performed on a limited number of patients ($n = <50$), proposed that in addition to cytokines and chemokines, a neutrophil activation signature, monocyte chemoattractants, pro-apoptotic factors, and HGF were associated with severe COVID-19[12,13]. Based on these studies, it is likely that the cytokine release syndrome may drive immune cell infiltration, lung epithelial and endothelial cells apoptosis[14], suboptimal T-cell function[15], multi-organ failure, and ultimately death[16].

Lymphocytopenia is also a hallmark of SARS-CoV2 infection and correlates with disease severity and death[17,18]. Indeed, patients with severe COVID-19 harbor a marked decrease in the absolute cell counts of T cells (both CD4 and CD8), B cells, and NK cells[18]. However, a number of studies have shown robust CD8 and/or CD4 T cell activation and proliferation compared to healthy controls in the majority of patients studied[19–21]. Virus-specific CD4 and CD8 T cell responses were predominantly directed against the M, Spike, and N proteins[21], tended to have a central memory phenotype (CD27+CD45RO+) and consisted of both mono and/or polyfunctional CD4 (IFNγ, IL-2, and TNF) and CD8 (IFN-γ, TNF, CD107a) T cells[22]. However, there was no correlation between the function of virus-specific CD4 and CD8 T cells and disease severity[22].

Recent studies have indicated that a deficient type I interferon (both IFN-α and β) response is associated with excessive inflammation and severe disease and about 10% of patients with anti-IFN antibodies experience severe disease requiring hospitalization in ICU[23,24].

In the present study, we investigated over 170 immunological parameters to identify signatures associated with the severity of COVID-19 at hospital admission. We studied three cohorts, one 'discovery' (LUH-1) and two 'validation' cohorts (FCS and LUH-2) including a total of 207 patients of which 85 were ICU patients and 122 non-ICU patients. We have identified two cytokines, i.e., HGF and CXCL13, as the best immunological signature predicting the severity of COVID-19 requiring ICU admission.

## Results

**Patient cohorts.** The aim of the present study was to define the immune-inflammatory profile of SARS-CoV-2 infection and to determine whether unique immune signatures may help identify patients with severe COVID-19 requiring ICU admission, referred to as ICU patients, versus those with moderate COVID-19 admitted in the internal medicine ward, referred to as non-ICU

patients. To achieve this objective, 98 adult patients with a PCR-confirmed SARS-CoV2 infection sequentially admitted to the Lausanne University Hospital were enrolled in a 'discovery' cohort (LUH-1) between 12 March and 4 April. Amongst the 98 patients, 43 were admitted directly to the ICU and 55 to the internal medicine ward. Blood and serum samples were collected at the time of admission and ex vivo cellular and serum immune signatures were determined using mass cytometry and multiplex beads assay. After the identification of immune signatures differentiating ICU from non-ICU patients in the 'discovery' cohort, the unique signatures were confirmed in 62 patients enrolled in the FCS cohort including 31 ICU and 31 non-ICU patients, and additional 47 patients in the LUH-2 cohort including 11 ICU and 36 non-ICU patients. The patients of the FCS and LUH-2 validation cohorts were enrolled between 25 January 2020 and 8 April 2020 and 7 April and 15 October, respectively, and the immunological profiles were analyzed blindly. Reference values for the immunological parameters investigated were derived from the analyses of a separate cohort of 450 healthy donors balanced for gender and age.

Demographic and clinical data of the patients enrolled in the 'discovery' cohort are summarized in Supplementary Table 1. Admission to the ICU for the LUH-1 followed the recommendations of the guidelines of the Swiss Federal Office of Public Health. This may explain the lack of difference for certain demographic parameters such as age and co-morbidities between ICU and non-ICU patients.

The most common symptoms included fever, cough, dyspnea, fatigue, myalgia/arthralgia, nausea/vomiting, and anosmia/dysgeusia (Supplementary Table 1). No significant differences in comorbidities were observed between non-ICU and ICU patients ($P > 0.05$). Complications were more frequently observed in ICU than in non-ICU patients ($P < 0.05$) including acute respiratory distress syndrome, community-acquired or hospital-acquired pneumonia, pulmonary embolism, septic shock, and acute hepatic injury (Supplementary Table 2).

The oxygen saturation was significantly lower in ICU patients than in non-ICU patients (95% *versus* 97%; $P < 0.05$), while the $FIO_2$ was significantly higher in ICU than in non-ICU patients (43% *versus* 21%; $P < 0.05$) (Supplementary Table 1). The total white cell blood count was significantly higher in ICU than in non-ICU patients (8.3 versus $6.7 \times 10^9$/Liter; $P < 0.05$) (Supplementary Table 1). Consistent with other studies[25], clinical parameters of inflammation such as C reactive protein (CRP), pro-calcitonin, and ferritin were markedly elevated and significantly higher in ICU than in non-ICU patients ($P < 0.003$) (Supplementary Table 1).

Finally, ICU patients were more frequently treated with tocilizumab, any antibiotic therapy, inhibitors of the renin–angiotensin–aldosterone system than non-ICU patients ($P < 0.001$) (Supplementary Table 3).

**Immune profile of circulating cell populations in ICU and non-ICU patients.** To determine the immune profile of ICU and non-ICU patients we investigated over 170 immunological parameters. We first assessed the influence of SARS-CoV2 infection on the absolute blood counts of CD4 and CD8 T-, B-, gamma-delta T-, NK, monocytic, and dendritic cell populations using a panel of 45 surface markers by mass cytometry (all gating strategies are available in Supplementary Fig. 1). Blood samples were collected from the 38 ICU and 53 non-ICU individuals enrolled in the 'discovery' cohort and compared to the reference normal value of 63 blood samples of healthy donors. ICU and non-ICU patients showed significant T cell lymphocytopenia ($P < 0.05$) (Supplementary Fig. 2). With regard to CD4 T cells, all

CD4 T cell populations were significantly reduced as compared to healthy donors ($P < 0.05$) (Supplementary Fig. 2a). CD8 T cells, total, naive, and effector memory (EM) cell populations were significantly reduced as compared to healthy donors ($P < 0.05$) while central memory (CM) CD8 T cells were significantly increased ($P < 0.001$) and terminally differentiated effector memory (TDEM) unchanged (Supplementary Fig. 2a). Consistent with a previous study[20], ICU and non-ICU patients had increased proportion of activated (HLA-DR$^+$CD38$^+$) memory (CD45RA$^-$CD27$^-$) CD4 and CD8 T cells as compared to healthy donors ($P < 0.001$) (Supplementary Fig. 3) while no significant differences were observed between ICU and non-ICU patients ($P > 0.05$) (Supplementary Fig. 3). PD-1 expression increased significantly only on memory (CD45RA$^-$CD27$^-$) CD4 and CD8 T cells in non-ICU patients ($P < 0.05$) (Supplementary Fig. 3).

The absolute total B cell number was not substantially influenced by SARS-CoV2 infection ($P > 0.05$) in both ICU and non-ICU patients. However, significant increases were observed in activated B cells (CD21$_{low}$CD38$_{low}$) and plasma cells (CD20$_{low}$CD38$^{high}$) whereas unswitched memory B cells (CD27$^-$IgD$^+$IgM$^+$), IgG2$^+$ switched memory B cells (CD27$^+$IgD$^-$IgM$^-$IgG2$^+$) and transitional B cells (CD38$^{high}$CD24$^{high}$IgM$^+$IgD$^+$CD10$^-$) were significantly reduced as compared to healthy individuals ($P < 0.05$) (Supplementary Fig. 2b).

Finally, both ICU and non-ICU patients showed a significant reduction in the cell number of gamma-delta T cells, plasmacytoid dendritic cells (DC) (pDC), myeloid, conventional, and inflammatory DC populations as compared to healthy individuals ($P < 0.001$) (Supplementary Fig. 2c). Except for gamma-delta T cells, all subsets of innate immune cell populations were more profoundly reduced in ICU patients than in non-ICU patients ($P < 0.05$ to $P < 0.001$).

The distribution of different CD4 T cell lineages and the phosphoprotein signaling profiles were then determined in CD4 T cell populations of 25 ICU and 50 non-ICU patients enrolled in the 'discovery' cohort and compared to blood samples of 146 healthy subjects using two mass cytometry panels composed of 43 and 37 markers. Cumulative data indicated that COVID-19 significantly influenced the distribution of blood CD4 T cell lineages. Indeed, the proportion of T helper type 1 (Th1) (CXCR3$^+$Tbet$^+$), Th17 cells (CCR6$^+$ROR$\gamma$t$^+$) and Tregs (CD25$^+$CD127$^-$FoxP3$^+$) were significantly increased at the expense of Th2 cells (CCR4$^+$Gata3$^+$) in both ICU and non-ICU patients as compared to healthy individuals ($P < 0.001$) (Fig. 1a). However, no significant differences were observed between ICU and non-ICU patients ($P > 0.05$) (Fig. 1a).

The ex vivo expression levels of phospho-STAT1 (pSTAT1), pSTAT3, and pSTAT5 were significantly increased in CD4 T cells in both ICU and non-ICU patients as compared to healthy individuals ($P < 0.001$) (Fig. 1b), suggesting a recent exposure to cytokines or growth factors[26]. Of note, several phosphorylated molecules such as pNF-κb, pCREB, pERK1/2, pS6, and p38, involved distinct signaling pathways, were increased but no significant difference was observed between ICU and non-ICU patients ($P > 0.05$) (Fig. 1b).

**Cytokine signatures in ICU versus non-ICU patients.** Recent studies have identified a number of markers potentially predictive of COVID-19 severity[11,12]. We determined whether a cytokine signature could help identifying at the time of hospital admission patients with severe COVID-19 requiring ICU admission. We, therefore, assessed the serum levels of a large panel ($n = 49$) of mediators including cytokines, soluble cytokine receptors, chemokines, and growth factors in blood samples collected at the time of admission in 43 ICU patients and 55 non-ICU patients

enrolled in the 'discovery' cohort (LUH-1). The serum concentration of these 49 markers of inflammation were compared to the levels measured in 450 sera collected from healthy individuals that were used as normal reference values (Fig. 2 and Supplementary Fig. 4). Serum levels of a large panel of cytokines, chemokines, and growth factors were markedly increased in ICU and non-ICU patients compared to those of healthy individuals ($P < 0.05$) (Fig. 2). However, serum levels of CCL4, CCL11, nerve growth factor-β (NGF-β), epidermal growth factor (EGF), fibroblast growth factor-2 (FGF-2) and placental growth factor-1 (PlGF-1) were significantly decreased in both ICU and non-ICU patients compared to healthy individuals ($P < 0.05$ to $P < 0.001$) (Fig. 2). Of note, serum levels of IL-1RA, IL-1β, IL-6, IL-10, IL-15, CCL2, CCL4, CXCL9, CXCL10, CXCL13, HGF, LIF, and VEGF-A were significantly increased in ICU versus non-ICU patients ($P < 0.001$) (Fig. 2).

To better define the serum factor signatures potentially differentiating ICU from non-ICU individuals, the levels of the 49 serum factors were compared between groups using Kruskal–Wallis test corrected for multiple comparisons. For each candidate marker, the optimal cutpoint to distinguish between ICU and non-ICU patients was determined using the cutpt command of Stata, applying the Liu method to maximize the product of the sensitivity and specificity. Based on the cutpoints, the candidate markers were dichotomized into lower and higher or equal to the cutpoint and the area under the receiver-operating curve (AUC), the sensitivity, specificity, positive and negative predictive values, and the likelihood ratio (Table 1) were computed. This analysis identified a panel of 13 serum factors (IL-10, CCL2, CCL4, CXCL13, IL-1RA, IL-6, IL-15, VEGF-A, CXCL9, CXCL10, IL-1β, LIF, and HGF) differently distributed between ICU and non-ICU patients (Supplementary Fig. 5). Based on these analyses, HGF and CXCL13 showed the best sensitivity (88.6% for both HGF and CXCL13) and specificity (81.5% for HGF and 79.6% for CXCL13) to discriminate between ICU and non-ICU patients (Table 1). More importantly, the positive predictive values (PPV) were 79.6% for HGF and 78% for CXCL13 and the negative predictive values (NPV) were 98.9% for HGF and 89.6% for CXCL13.

We then performed a blinded evaluation of the serum levels of the 49 cytokines in samples collected from patients enrolled in two independent 'validation' COVID-19 cohorts of the FCS ($n = 62$ patients) and of the LUH-2 cohort ($n = 47$ patients). The LUH-2 cohort was enrolled based on the same criteria of the LUH-1 cohort. Demographic and clinical data of the FCS 'validation' cohort are summarized in Supplementary Table 4. Admission to the ICU for the FCS followed the recommendations of the guidelines of the French Haute Autorité de Santé. We then applied the cutpoints values for the 13 serum factors (IL-10, CCL2, CCL4, CXCL13, IL-1RA, IL-6, IL-15, VEGF-A, CXCL9, LIF, IL-1β, CXCL10, and HGF) defined in the 'discovery' cohort. Following unblinding of the FCS, increased levels of HGF and CXCL13 predicted ICU admission in 27 (87.0%) of 31 patients and non-ICU admission in 29 (93.5%) of 31 patients. Following unblinding of the LUH-2 cohort, ICU admission was predicted in 34 (94.4%) of 36 patients and internal medicine ward admission in 10 (90.9%) of 11 patients. ROC and AUC analyses confirmed the hierarchy amongst the 13 selected cytokines in discriminating between ICU and non-ICU patients in the FCS and LUH-2 validation cohorts (Table 2).

Thus, HGF and CXCL13 were the best predictors of COVID-19 severity and ICU admission. Interestingly, the combination of HGF and CXCL13 further improved their discriminative power for ICU admission in the 'discovery' and 'validation' cohorts (Table 3). The performance of the combination of the two

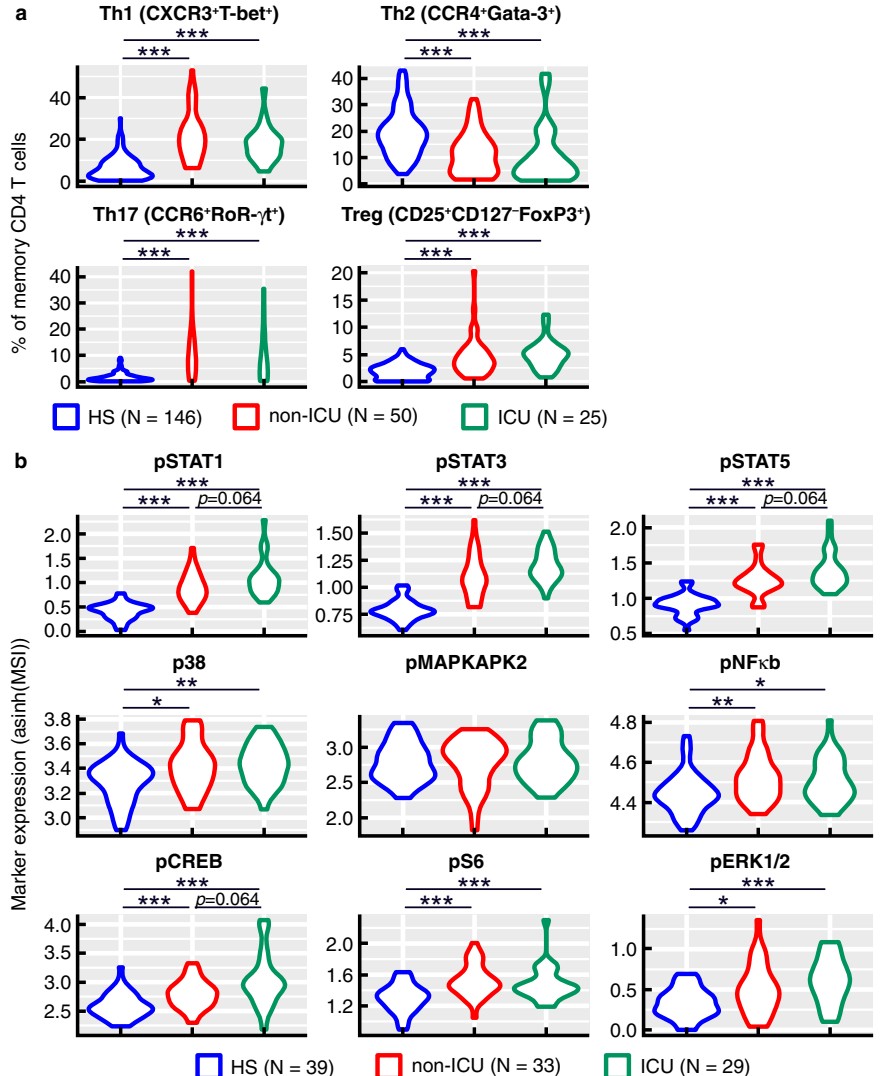

**Fig. 1 Distribution of CD4 T cell lineage and phosphoprotein signaling profiles in non-ICU and ICU COVID-19 patients. a** Frequencies of Th1 (CXCR3 +T-bet+), Th2 (CCR4+Gata-3+), Th17 (CCR6+RoR-γt+) and Treg (CD25+CD127-FoxP3+) CD4 T cell sub-populations in healthy subjects ($N = 146$), non-ICU ($N = 50$) and ICU ($N = 25$) patients. **b** Mean signal intensity of ex vivo phospho-STAT1 (pSTAT1), pSTAT3, pSTAT5, p38, pMAPKAP2, pNFkB, pCREB, pS6 and pERK1/2 in healthy subjects ($N = 39$), non-ICU ($N = 33$) and ICU ($N = 29$) patients. Blue plots correspond to healthy subjects (H.S), red plots correspond to non-ICU patients and green plots correspond to ICU patients. Black stars indicate statistical significance between ICU or non-ICU patients and healthy subjects. Statistical significance ($P$ values) was obtained using two-sided Kruskal–Wallis test, using a Bonferroni correction. *$P < 0.05$; **$P < 0.01$; ***$P < 0.001$. Exact $P$ values are available in Source Data file.

cytokines in the 'discovery' cohort in the France COVID-19 Study 'validation' cohort are shown in Table 3.

We next assessed the potential of the 13 serum factors (IL-10, CCL2, CCL4, CXCL13, IL-1RA, IL-6, IL-15, VEGF-A, CXCL9, LIF, IL-1β, CXCL10, and HGF) and their relative cutpoint values to predict 30-day mortality among the COVID-19 patients enrolled in the combined LUH-1, LUH-2, and FCS cohorts. Among the initial 207 patients, vital status at 30 days was available for 197 and 186 had data allowing for survival analysis. The associations between categories of markers and vital status were assessed by chi-square; survival analysis was performed via a multilevel survival model using a Weibull distribution and results were expressed as multivariable-adjusted hazards ratio (HR) with a 95% confident interval (CI). Overall, 18 patients died, 17 of whom had high levels of the combination of HGF and CXCL13 ($P = 0.006$); survival analysis showed that patients with the combination of HGF and CXCL13 had a 8.80-fold higher likelihood of dying ($P = 0.054$) (Table 4).

## Discussion

The hallmark of severe COVID-19 is an acute respiratory distress syndrome (ARDS) with respiratory failure requiring mechanical ventilation in 10–24% of hospitalized patients. A large number of studies have drawn attention to systemic immune activation involving both the innate and adaptive arms of the host immune system[11,12,20,27,28]. The relevance in COVID-19 of the massive release of a large number of soluble mediators including cytokines, cytokine receptors, growth factors, and chemokines has been thoroughly discussed in a recent 'Opinion' article[29]. The article has highlighted that the pathophysiology of the COVID-19 cannot be explained solely on the basis of the increase in a few inflammatory cytokines such as IL-6 and TNF. It remains unclear to what extent the increase of circulating mediators drives the pathogenesis of severe COVID-19.

A large number of studies have been carried out to better understand the pathophysiology of COVID-19 and identify predictive markers of disease severity in the early symptomatic phase

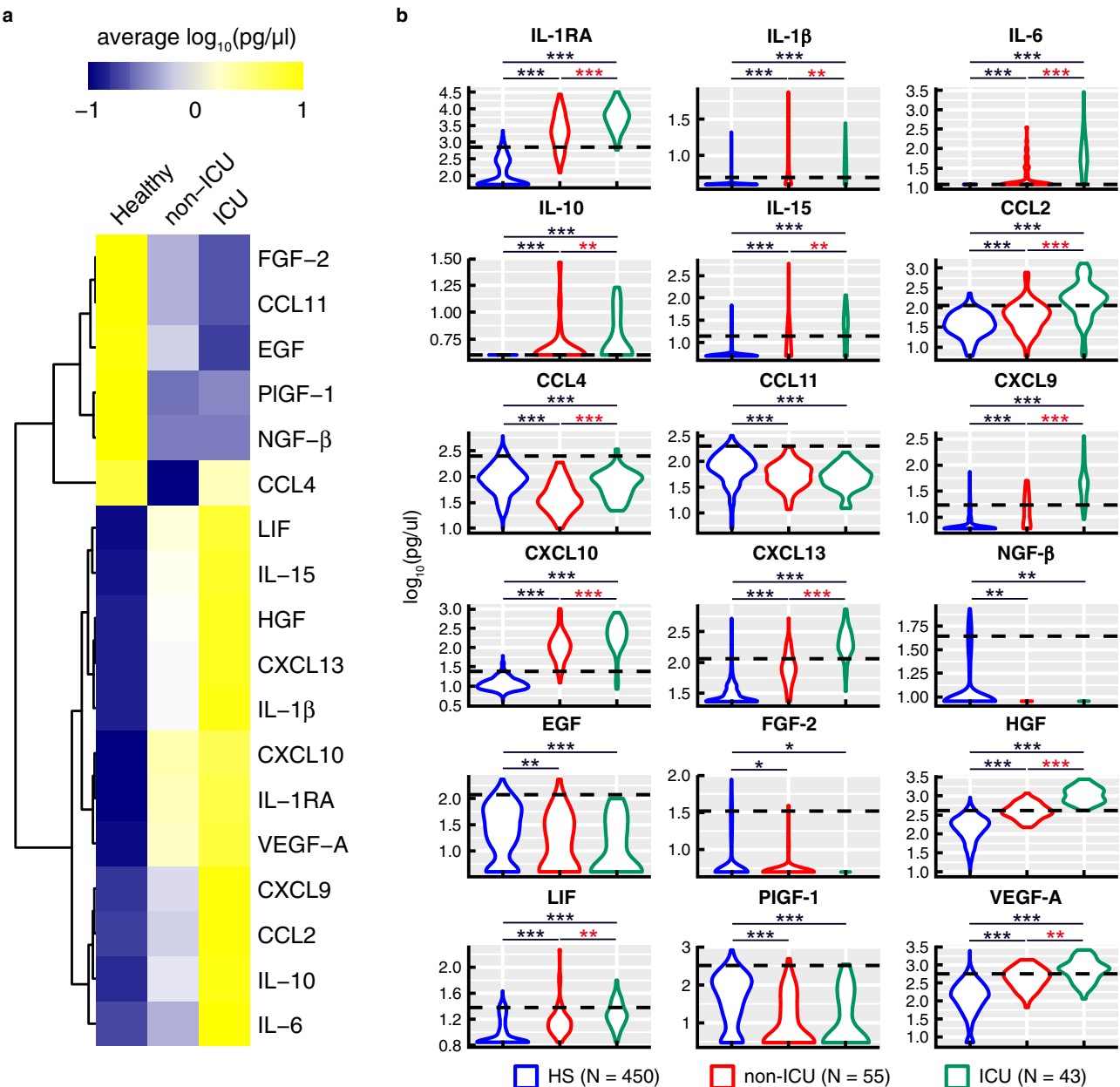

**Fig. 2 Serum cytokine, soluble cytokine receptor, chemokine, and growth factor profiles in non-ICU and ICU COVID-19 patients. a** Heat-map representing the mean serum cytokine levels detected in healthy subjects ($N = 450$), non-ICU ($N = 55$) and ICU ($N = 43$) patients. Blue-to-yellow color code represents low-to-high average cytokine levels. Cytokine level similarities are represented by a dendrogram constructed by hierarchical clustering. **b** Levels of cytokines (IL-1β, IL-6, IL-10, and IL-15), cytokine receptor (IL-1RA), chemokines (CCL2, CCL4, CCL11, CXCL9, CXCL10, and CXCL13) and growth factors (NGF-β, EGF, HGH, LIF, PIGF-1, and VEGF-A) in healthy subjects ($N = 450$), non-ICU ($N = 55$) and ICU ($N = 43$) patients. Blue plots correspond to healthy subjects (HS), red plots corresponds to non-ICU patients and green plots correspond to ICU patients. Dotted line represents the upper normal values. Black stars indicate statistical significance between ICU or non-ICU patients and healthy subjects. Statistical significance (*P* values) was obtained using two-sided Kruskal–Wallis test, using a Bonferroni correction. *$P < 0.05$; **$P < 0.01$; ***$P < 0.001$. Exact *P* values are available in Source Data file.

of infection[11,12,20,27,28]. Consistent with these studies[11,12,27,28], we observed that several cellular markers of activation and differentiation of blood T, B, monocyte, and DC cell populations were abnormal in SARS-CoV2 infected patients compared to healthy individuals. However, none of these cellular markers can discriminate between severe and moderate COVID-19. Of note, we have also shown in SARS-CoV2 patients an increase of Th1 and Th1/Th17 CD4 T cell lineages and a decrease in Th2 cells supporting the inflammatory profile of the T cell response associated with COVID-19. Furthermore, the increase in signaling pathways such as pNF-κb, pCREB, pERK1/2, pS6, and p38 is consistent

with the cytokine-mediated activation of the different pro-inflammatory CD4 T cell lineages.

Consistent with the previous studies[11,12,28], we confirmed the increase in a large number of soluble mediators in patients with COVID-19 as compared to the values obtained in samples collected from healthy individuals. However, the results of the present study provide a substantial advance in the understanding of the pathophysiology of COVID-19 and in the identification of predictive markers of the severity of the SARS-CoV2 infection. Two recent studies attempted to identify markers of disease severity. In one study[11], only a small number ($n = 4$) of cytokines were measured

**Table 1 Performance (area under the receiver-operating curve (AUC), sensitivity, specificity, positive and negative predictive values, and likelihood ratio) of each candidate markers dichotomized into lower than, higher, or equal to the cutpoint distinguishing ICU and non-ICU COVID-19 patients of the discovery cohort.**

| Marker | Cutpoint | AUC | Sensitivity | Specificity | Positive predictive value | Negative predictive value | Likelihood ratio |
|---|---|---|---|---|---|---|---|
| HGF | 593.1 | 0.911 (0.854-0.969) | 88.6 (75.4-96.2) | 81.5 (68.6-90.7) | 79.6 (65.7-89.8) | 98.9 (98.6-99.2) | 4.79 (2.71-8.46) |
| CXCL13 | 119.7 | 0.875 (0.801-0.948) | 88.6 (75.4-96.2) | 79.6 (66.5-89.4) | 78.0 (64.0-88.5) | 89.6 (77.3-96.5) | 4.35 (2.54-7.45) |
| CXCL9 | 19 | 0.869 (0.801-0.936) | 84.1 (69.9-93.4) | 74.1 (60.3-85.0) | 72.5 (58.3-84.1) | 85.1 (71.7-93.8) | 3.24 (2.03-5.18) |
| IL-6 | 25.6 | 0.796 (0.708-0.884) | 77.3 (62.2-88.5) | 79.6 (66.5-89.4) | 75.6 (60.5-87.1) | 81.1 (68.0-90.6) | 3.79 (2.19-6.58) |
| CCL2 | 121.3 | 0.775 (0.676-0.873) | 65.9 (50.1-79.5) | 85.2 (72.9-93.4) | 78.4 (61.8-90.2) | 75.4 (62.7-85.5) | 4.45 (2.27-8.73) |
| CXCL10 | 156 | 0.743 (0.640-0.846) | 68.2 (52.4-81.4) | 75.9 (62.4-86.5) | 69.8 (53.9-82.8) | 74.5 (61.0-85.3) | 2.83 (1.69-4.74) |
| IL-1RA | 2741.7 | 0.734 (0.635-0.833) | 79.5 (64.7-90.2) | 61.1 (46.9-74.1) | 62.5 (48.5-75.1) | 78.6 (63.2-89.7) | 2.05 (1.42-2.95) |
| CCL4 | 40 | 0.714 (0.611-0.818) | 79.5 (64.7-90.2) | 59.3 (45.0-72.4) | 61.4 (47.6-74.0) | 78.0 (62.4-89.4) | 1.95 (1.37-2.78) |
| VEGF-A | 677.4 | 0.702 (0.597-0.807) | 61.4 (45.5-75.6) | 77.8 (64.4-88.0) | 69.2 (52.4-83.0) | 71.2 (57.9-82.2) | 2.76 (1.59-4.79) |
| IL-15 | 16.2 | 0.689 (0.581-0.796) | 65.9 (50.1-79.5) | 72.2 (58.4-83.5) | 65.9 (50.1-79.5) | 72.2 (58.4-83.5) | 2.37 (1.47-3.83) |
| IL-10 | 3.1 | 0.687 (0.599-0.775) | 50.0 (34.6-65.4) | 88.9 (77.4-95.8) | 78.6 (59.0-91.7) | 68.6 (56.4-79.1) | 4.50 (2.00-10.1) |
| IL-1β | 4.325 | 0.690 (0.582-0.797) | 69.8 (53.9-82.8) | 70.9 (57.1-82.4) | 65.2 (49.8-78.6) | 75 (61.1-86.0) | 2.4 (1.5-3.8) |
| LIF | 15.23 | 0.703 (0.597-0.809) | 65.1 (49.1-79.0) | 72.7 (59.0-83.9) | 65.1 (49.1-79.0) | 72.7 (59.0-83.9) | 2.4 (1.5-3.9) |

**Table 2 Performance (AUC, sensitivity, specificity, positive and negative predictive values, and likelihood ratio) of each candidate markers dichotomized into lower than, higher or equal to the cutpoint distinguishing ICU and non-ICU COVID-19 patients in FCS and LUH-2 validation cohorts.**

| Marker | Cutpoint | AUC | Sensitivity | Specificity | Positive predictive value | Negative predictive value | Likelihood ratio |
|---|---|---|---|---|---|---|---|
| *FCS validation cohort* | | | | | | | |
| HGF | 593.1 | 0.976 (0.948-1.000) | 87.1 (70.2-96.4) | 93.5 (78.6-99.2) | 93.1 (77.2-99.2) | 87.9 (71.8-96.6) | 13.5 (3.5-51.9) |
| CXCL13 | 119.7 | 0.903 (0.832-0.974) | 96.8 (83.3-99.9) | 58.1 (39.1-75.5) | 69.8 (53.9-82.8) | 94.7 (74.0-99.9) | 2.3 (1.5-3.5) |
| CXCL9 | 19.0 | 0.814 (0.706-0.921) | 51.6 (33.1-69.8) | 87.1 (70.2-96.4) | 80.0 (56.3-94.3) | 64.3 (48.0-78.4) | 4.0 (1.5-10.6) |
| IL-6 | 25.6 | 0.661 (0.578-0.745) | 16.1 (5.5-33.7) | 100 (88.8-100) | 100 (47.8-100) | 54.4 (40.7-67.6) | Not computable |
| CCL2 | 121.3 | 0.745 (0.622-0.867) | 29.0 (14.2-48.0) | 100 (88.8-100) | 100 (66.4-100) | 58.5 (44.1-71.9) | Not computable |
| CXCL10 | 156.0 | 0.782 (0.663-0.901) | 25.8 (11.9-44.6) | 93.5 (78.6-99.2) | 80.0 (44.4-97.5) | 55.8 (41.3-69.5) | 4.0 (0.9-17.4) |
| IL-1RA | 2741.7 | 0.803 (0.692-0.915) | 35.5 (19.2-54.6) | 96.8 (83.3-99.9) | 91.7 (61.5-99.8) | 60.0 (45.2-73.6) | 11.0 (1.5-80.1) |
| CCL4 | 40.0 | 0.714 (0.581-0.846) | 100 (88.8-100) | 9.7 (2.0-25.8) | 52.5 (39.1-65.7) | 100 (29.2-100) | 1.1 (1.0-1.2) |
| VEGF-A | 677.4 | 0.856 (0.765-0.947) | 83.9 (66.3-94.5) | 58.1 (39.1-75.5) | 66.7 (49.8-80.9) | 78.3 (56.3-92.5) | 2.0 (1.3-3.1) |
| IL-15 | 16.2 | 0.766 (0.647-0.885) | 38.7 (21.8-57.8) | 90.3 (74.2-98.0) | 80.0 (51.9-95.7) | 59.6 (44.3-73.6) | 4.0 (1.3-12.8) |
| IL-10 | 3.1 | 0.594 (0.512-0.676) | 22.6 (9.6-41.1) | 96.8 (83.3-99.9) | 87.5 (47.3-99.7) | 55.6 (41.4-69.1) | 7.0 (0.9-53.6) |
| IL-1β | 4.325 | 0.604 (0.473-0.734) | 38.7 (21.8-57.8) | 77.4 (58.9-90.4) | 63.2 (38.4-83.7) | 55.8 (39.9-70.9) | 1.7 (0.8-3.8) |
| LIF | 15.23 | 0.652 (0.540-0.765) | 16.1 (5.5-33.7) | 96.8 (83.3-99.9) | 83.3 (35.9-99.6) | 53.6 (39.7-67.0) | 5.0 (0.6-40.4) |
| *LUH-2 validation cohort* | | | | | | | |
| HGF | 593.1 | 0.976 (0.948-1.000) | 87.1 (70.2-96.4) | 93.5 (78.6-99.2) | 93.1 (77.2-99.2) | 87.9 (71.8-96.6) | 13.5 (3.5-51.9) |
| CXCL13 | 119.7 | 0.903 (0.832-0.974) | 96.8 (83.3-99.9) | 58.1 (39.1-75.5) | 69.8 (53.9-82.8) | 94.7 (74.0-99.9) | 2.3 (1.5-3.5) |
| CXCL9 | 19.0 | 0.814 (0.706-0.921) | 51.6 (33.1-69.8) | 87.1 (70.2-96.4) | 80.0 (56.3-94.3) | 64.3 (48.0-78.4) | 4.0 (1.5-10.6) |
| IL-6 | 25.6 | 0.661 (0.578-0.745) | 16.1 (5.5-33.7) | 100 (88.8-100) | 100 (47.8-100) | 54.4 (40.7-67.6) | Not computable |
| CCL2 | 121.3 | 0.745 (0.622-0.867) | 29.0 (14.2-48.0) | 100 (88.8-100) | 100 (66.4-100) | 58.5 (44.1-71.9) | Not computable |
| CXCL10 | 156.0 | 0.782 (0.663-0.901) | 25.8 (11.9-44.6) | 93.5 (78.6-99.2) | 80.0 (44.4-97.5) | 55.8 (41.3-69.5) | 4.0 (0.9-17.4) |
| IL-1RA | 2741.7 | 0.803 (0.692-0.915) | 35.5 (19.2-54.6) | 96.8 (83.3-99.9) | 91.7 (61.5-99.8) | 60.0 (45.2-73.6) | 11.0 (1.5-80.1) |
| CCL4 | 40.0 | 0.714 (0.581-0.846) | 100 (88.8-100) | 9.7 (2.0-25.8) | 52.5 (39.1-65.7) | 100 (29.2-100) | 1.1 (1.0-1.2) |
| VEGF-A | 677.4 | 0.856 (0.765-0.947) | 83.9 (66.3-94.5) | 58.1 (39.1-75.5) | 66.7 (49.8-80.9) | 78.3 (56.3-92.5) | 2.0 (1.3-3.1) |
| IL-15 | 16.2 | 0.766 (0.647-0.885) | 38.7 (21.8-57.8) | 90.3 (74.2-98.0) | 80.0 (51.9-95.7) | 59.6 (44.3-73.6) | 4.0 (1.3-12.8) |
| IL-10 | 3.1 | 0.594 (0.512-0.676) | 22.6 (9.6-41.1) | 96.8 (83.3-99.9) | 87.5 (47.3-99.7) | 55.6 (41.4-69.1) | 7.0 (0.9-53.6) |
| IL-1β | 4.325 | 0.604 (0.473-0.734) | 38.7 (21.8-57.8) | 77.4 (58.9-90.4) | 63.2 (38.4-83.7) | 55.8 (39.9-70.9) | 1.7 (0.8-3.8) |
| LIF | 15.23 | 0.652 (0.540-0.765) | 16.1 (5.5-33.7) | 96.8 (83.3-99.9) | 83.3 (35.9-99.6) | 53.6 (39.7-67.0) | 5.0 (0.6-40.4) |

**Table 3 Performance (sensitivity, specificity, positive and negative predictive values, and the likelihood ratio) of the combination of HGF and CXCL13 to further improve the discrimination between ICU and non-ICU COVID-19 patients.**

| Cohort | N | Sensitivity | Specificity | Positive predictive value | Negative predictive value | Likelihood ratio |
|---|---|---|---|---|---|---|
| LUH-1 | 98 | 79.1 (64.0-90.0) | 94.5 (84.9-98.9) | 91.9 (78.1-98.3) | 85.2 (73.8-93.0) | 14.5 (4.8-44.0) |
| LUH-2 | 47 | 81.8 (48.2-97.7) | 94.4 (81.3-99.3) | 81.8 (48.2-97.7) | 94.4 (81.3-99.3) | 14.7 (3.7-58.3) |
| FCS | 62 | 87.1 (70.2-96.4) | 93.5 (78.6-99.2) | 93.1 (77.2-99.2) | 87.9 (71.8-96.6) | 13.5 (3.5-51.9) |

and IL-6, TNF, and IL-8 were identified as markers of severity of COVID-19 as measured by mortality. The study was conducted in a large number of patients but was unable to predict the severity of the disease at the time of hospital admission. In two studies, conducted on a small number ($n = 49$[12] and $n = 40$[13]) of patients,

HGF in addition to other markers was proposed to serve as a marker of severity of COVID-19.

It is important to underscore that in our study the serum samples were collected at the time of hospital admission in the 'discovery' and in the two 'validation' cohorts. The timing of

**Table 4 Performance of each candidate markers dichotomized into lower than or higher than the cutpoint or of the combination of HGF and CXCL13 to predict death during the follow-up of COVID-19 patients enrolled in LUH-1, LUH-2 and the FCS cohorts.**

| Marker | Low | High | p-value‡ | Hazard ratio* | p-value‖ |
|---|---|---|---|---|---|
| HGF | 5 (4.6) | 13 (14.9) | 0.012 | 1.53 (0.29–8.18) | 0.621 |
| CXCL13 | 2 (2.4) | 16 (14.0) | 0.005 | 4.94 (0.85–28.6) | 0.075 |
| CXCL9 | 5 (4.6) | 13 (14.6) | 0.016 | 1.02 (0.32–3.26) | 0.980 |
| IL-6 | 10 (7.1) | 8 (14.3) | 0.114 | 1.33 (0.45–3.87) | 0.606 |
| CCL2 | 12 (8.1) | 6 (12.5) | 0.352 | 0.66 (0.21–2.03) | 0.463 |
| CXCL10 | 9 (6.7) | 9 (14.5) | 0.076 | 3.73 (1.14–12.2) | 0.029 |
| IL-1RA | 8 (6.3) | 10 (14.3) | 0.063 | 2.39 (0.73–7.82) | 0.151 |
| CCL4 | 2 (4.6) | 16 (10.5) | 0.230 | 2.57 (0.48–13.7) | 0.269 |
| VEGF-A | 8 (8.0) | 10 (10.3) | 0.574 | 1.23 (0.40–3.74) | 0.721 |
| IL-15 | 11 (8.7) | 7 (9.9) | 0.792 | 0.85 (0.28–2.58) | 0.780 |
| IL-10 | 13 (8.5) | 5 (11.4) | 0.561 | 0.81 (0.26–2.50) | 0.712 |
| IL-1β | 12 (10.1) | 6 (7.7) | 0.569 | 0.45 (0.15–1.36) | 0.158 |
| LIF | 12 (8.1) | 6 (12.2) | 0.384 | 0.74 (0.24–2.26) | 0.597 |
| *Combination of HGF and CXCL13* | | | | | |
| HGF/CXCL13 | 1 (1.5) | 17 (13.3) | 0.006 | 8.80 (0.96–80.3) | 0.054 |

The first two columns indicate the percentage of subjects within a given category (low or high levels) who died during follow-up, all cohorts together.
*Adjusted for age (continuous), ICU stay (yes/no) and cohort (Lausanne 1/Lausanne 2/Paris), ‡analysis by chi-square; ‖, analysis by a multilevel survival model using a Weibull distribution, where patients were nested within each cohort.

sampling is critical because serum cytokine levels can change substantially as the infection progresses. We have shown that, among the 49 soluble mediators measured, two cytokines, HGF and CXCL13, are the best predictors of the need for ICU hospitalization for COVID-19 patients.

HGF is a pleiotropic cytokine produced by mesenchymal cells and macrophages. It is required for normal embryogenesis and development[30,31] of several organs including the lung[32]. In adults, HGF is produced following injury of the lung tissue and promotes tissue repair[33–36]. HGF promotes lung tissue repair through the inhibition of apoptosis of lung epithelial and endothelial cells, and by counteracting a number of pro-apoptotic and pulmonary fibrosis factors such as TGF-β, IL-1β, IL-8, TNF-α, the basic fibroblastic factor, the insulin-like growth factor, and the platelet-derived growth factor[37–46]. It has been proposed that the anti-apoptotic activity of HGF is due in particular to the activation of three signaling pathways, i.e., ERK/MAPK, PI3K/Akt, and STAT3[47–49].

HGF may play also a central role in the regulation of inflammation. A number of pro-inflammatory cytokines such as IFN-γ, IL-1α/β, and TNF-α induce HGF expression as well as activated T cells[50,51] while glucocorticoids and TGF-β inhibit HGF production[52]. HGF may induce monocyte-macrophage activation[53], B cell homing[54], and modulation of DC functions[55]. HGF exerts predominantly an anti-inflammatory role through the decrease production of IL-6 and increase production of IL-10[56,57], by preventing the differentiation of inflammatory T cell lineages through the suppression of DC-mediated IL-12p70 production[57,58], and by favoring Tregs maturation[57,59]. Finally, HGF produced by follicular DC is a positive regulator of growth and survival of B cells and plasma cells[51,60].

CXCL13 plays a central physiological role in the organization of secondary lymphoid tissue structure of primary and secondary follicles and thus of B cell maturation[61]. CXCL13 is a pro-inflammatory cytokine involved in several pathological conditions and the finding of increased levels in tissue and/or in serum corresponds to varying degrees of inflammation. CXCL13 serum levels have been found increased in several uncontrolled infectious disease such as in viremic HIV infection, in a variety of autoimmune diseases, and in both hematological and solid tumors (reviewed in ref. [61]). Interestingly, increased serum levels and tissue expression of CXCL13 have been initially found to be associated with idiopathic pulmonary fibrosis[62,63] and recently in several interstitial lung diseases including idiopathic interstitial pneumonia and interstitial pneumonia with autoimmune features[64]. The increased levels of CXCL13 are associated with severe prognosis and increased mortality in all the interstitial lung diseases. Furthermore, the CXCL13/CXCR5 axis (CXCL13 being the ligand of CXCR5) uses some of the signaling pathways such as ERK/MAPK and PI3/AKT (reviewed in ref. [61]).

Based on the biology of HGF, our observation of increased serum levels early in symptomatic infection and its association with ICU hospitalization is likely an indicator of an ongoing severe respiratory syndrome associated with interstitial pneumonia. Upregulation of HGF is the host's physiological counter-regulatory immune response to reduce inflammation, to limit lung tissue injury and to promote tissue repair. Consistent with this view, over 90% of non-ICU patients with a moderate respiratory syndrome had low levels of HGF. Of note, HGF may exert its anti-inflammatory property through IL-10. Interestingly, IL-10 was one of the thirteen cytokines found to discriminate ICU from non-ICU patients.

Interestingly, HGF has also been shown to be significantly increased in patients with severe influenza A (H1N1) virus infection[65] and in patients with inflammatory lung diseases (interstitial pneumonitis or bacterial pneumonia)[66]. Levels of HGF remained elevated over time and were more elevated in non-survivors as compared to survivors of acute lung injuries[66,67]. These studies highlighted the potential benefit of using HGF levels as a prognosis marker of inflammatory pulmonary diseases[66,67].

With regard to CXCL13, the early increase in the symptomatic severe COVID-19 may also reflect the potent host immune response to promote maturation of B cell and antibody response in order to achieve rapid control of the virus replication and virus clearance. However, the persistence of elevated levels of CXCL13 in the lung tissue and serum may be detrimental and responsible for fueling the inflammation and promoting lung fibrosis.

Of note, we have demonstrated that the combined use of HGF and CXCL13 provides a powerful immune signature discriminating between ICU and non-ICU patients at hospital admission with positive and negative predictive values ranging from 81.8 to 93.1% and 85.2 to 94.4% in the 3 cohorts, and predicting the occurrence of death during patient follow-up.

Therefore, the combined assessment of the two cytokines is a valuable tool in the clinical management of patients with acute SARS-CoV-2 infection.

In conclusion, the present study provides insights in the early pathophysiological events associated with severe COVID-19 and identified HGF and CXCL13 as critical pathogenic biomarkers of disease severity and best predictors of ICU admission and death.

## Methods

**Study group, ethics statement.** Eighty-eight ICU and one hundred twenty-five non-ICU hospitalized PCR-confirmed SARS-CoV2 infected individuals were enrolled in the present study. No statistical method was used to predetermine sample size. The sample size was estimated based on a previously published study[27]. The present study was approved by the ethical commission (CER-VD) and all subjects provided a written informed consent. As inclusion criteria, only patients with a positive SARS-CoV2 PCR were enrolled. Admission to ICU or to internal medicine ward (non-ICU) were the following: individuals with severe COVID-19 with acute respiratory failure requiring mechanical ventilation and/or cardio-circulatory insufficiency requiring the administration of vasoactive agents were admitted to ICU. Individuals with severe COVID-19 with acute respiratory failure requiring supplemental oxygen and did not have criteria for ICU admission were admitted to the internal medicine ward (non-ICU) required.

As exclusion criteria, pregnant women were not enrolled. Serum and blood samples were also collected from 450 healthy individuals during the pre-pandemic period. The exclusion criteria were sign of acute or chronic viral hepatitis (HAV, HBV, HCV, and HEV), prior diagnosis of autoimmune disease (e.g., rheumatoid arthritis, psoriasis, SLE), prior diagnosis of primary or secondary immunodeficiency (e.g., HIV infection), and current or past (last 4 weeks) use of medications that are known to modify the immune response.

**Assessment of serum immune signatures.** Serum concentration of cytokines and soluble cytokine receptors *i.e.* IL-1α, IL-1RA, IL-1β, IL-2, IL-4, IL-5, IL-6, IL-7, IL-9, IL-10, IL-12p70, IL-13, IL-15, IL-17A, IL-18, IL-21, IL -22, IL-23, IL-27, IL-31, IFN-α, IFN-γ and TNF, chemokines, i.e. CCL2, CCL3, CCL4, CCL5, CCL11, CXCL1, CXCL8, CXCL9, CXCL10, CXCL12, CXCL13 and TNF-β and growth factors, i.e., NGF-β, BDNF, EGF, FGF-2, HGF, LIF, PDGF-BB, PlGF-1, SCF, VEGF-A, VEGF-D, BAFF, GM-CSF and G-CSF were determined by multiplex bead assay as previously described[68]. The upper normal values for each marker were defined based on the results obtained in the 450 sera collected from healthy individuals (mean + 2 standard deviations).

**Immune profiling of circulating cell populations by mass cytometry.** Blood samples (200 μl) were first incubated (30 min; RT) with metal-conjugated antibodies directed against CD3, CD7, CD45, CCR7, CXCR3, CXCR5, and γδ TCR (c.f. antibodies section; Panel 1; Supplementary Data 1). Cells were then fixed (5 min; RT) with PBS 2.4% PFA and lysed (15 min, RT) using Bulklysis solution (Cytognos) and washed (PBS, 0.5% BSA, Sodium azide 0.02%). Cells were then incubated (30 min; RT) with the remaining metal-conjugated monoclonal antibodies (c.f. antibodies section). Cells were then washed (PBS, 0.5% BSA, Sodium azide 0.02%) and fixed (5 min; RT) with PBS 2.4% PFA. Cells were stained (1 h; RT) with DNA intercalator (1 μM Cell-ID Intercalator, Fluidigm/DVS Science) in PBS, 0.5% BSA, sodium azide 0.02%, 0.3% saponin, 1.6% PFA. The absolute counts of blood cell populations of ICU and non-ICU individuals were compared to blood samples collected from healthy individuals (c.f. Study group section).

**Evaluation of CD4 T cell lineage distribution by mass cytometry.** Blood samples (100 μl) were first incubated (30 min; RT) with metal-conjugated antibodies directed against CD8, CD4, CCR4, CD127, CCR6, CXCR3, CCR9, CCR7, CXCR5, CCR5 and CD45 (c.f. antibodies section; Supplementary material). Cells were then fixed (5 min; RT) with PBS 2% PFA and lysed (15 min, RT) using Bulklysis solution (Cytognos) and washed (PBS, 0.5% BSA, 0.02% Sodium azide). Cells were then incubated (30 min; RT) with the metal-conjugated monoclonal antibodies directed against CD3, CD44, CD25, CCR6, CXCR5, CD38, TIGIT, 2B4, PD1, CD27, CD69, CD45RO, CD127, CD16, CD31, CD95, CD57, NKG2D, CD45RA, HLA-DR, PD-L1, CD151, CD40L, ICOS, LAG3, OX40 (c.f. antibodies section; Panel 2; Supplementary Table 5 and Supplementary Data 1). Cells were then washed (PBS, 0.5% BSA, 0.02% Sodium azide) and fixed (5 min; RT) with PBS 2.4% PFA. Cells were then permeabilized (30 min; 4 °C) (Foxp3 Fixation/Permeabilization Kit; eBioscience) then washed and stained (30 min; 4 °C) with the metal-conjugated monoclonal antibodies directed against Tbet, Ki67, Bcl2, Rorγt, Gata3, FoxP3 (c.f. antibodies section; Panel 2; Supplementary Table 5 and Supplementary Data 1). Cells were then washed (PBS, 0.5% BSA, 0.3% saponin, 0.02% Sodium azide). Cells were stained (1 h; RT) with DNA intercalator (1 μM Cell-ID Intercalator, Fluidigm/DVS Science) in PBS, 0.5% BSA, 0.02% Sodium azide, 0.3% saponin, 1.6% PFA. The distribution of CD4 T cell lineages evaluated in ICU and non-ICU individuals were compared to values obtained from healthy individuals (c.f. Study group section).

**Assessment of the CD4 T cell phospho-protein signaling profile by mass cytometry.** Blood samples (200 μl) were barcoded using a strategy based on mass-tag (105 Pd, 104 Pd, 106 Pd, 108 Pd, and 110 Pd) palladium (Trace Sciences; 400 nM; 30 min; RT) and isotope-labeled (89Y, 111 Cd, 114 Cd, 116 Cd, 141Pr and 198Pt) anti-CD45 MAbs (HI30; 30 min; RT). Briefly, cells were stained with specific anti-CD45 MAbs and palladium mass-tag compound, then fixed (5 min; RT) with PBS 2.4% PFA and lysed (15 min, RT) using Bulklysis solution (Cytognos) and washed (PBS, 0.5% BSA, 0.02% Sodium azide). Cells were then pooled and incubated (30 min; RT) with the metal-conjugated monoclonal antibodies directed against CD3, CD45, CD8, CD4, CD19, CD1c, CD69, CD31, CD86, CD7, CD39, CD56, CD123, CD21, CD27, CD14, CD11c, CD62L, CD161, CD20, CD38, CD45RA, CD15, CD141, HLA-DR, CD57 and CD16 (c.f. antibodies section; Panel 3; Supplementary Table 5 and Supplementary Data 1). Cells were then washed (PBS, 0.5% BSA, 0.02% Sodium azide) and fixed (5 min; RT) with PBS 2.4% PFA. Cells were then permeabilized (30 min; 4 °C) (Foxp3 Fixation/Permeabilization Kit; eBioscience) then washed and stained (30 min; 4 °C) with the metal-conjugated monoclonal antibodies directed against pSTAT1, pSTAT3, pSTAT5, p38, pMAP-KAPK2, pNFkb, Ki67, pERK1/2, pS6, pCREB, (c.f. antibodies section; Panel 3; Supplementary Table 5 and Supplementary Data 1). Cells were then washed (PBS, 0.5% BSA, 0.3% saponin, 0.02% Sodium azide). Cells were stained (1 h; RT) with DNA intercalator (1 μM Cell-ID Intercalator, Fluidigm/DVS Science) in PBS, 0.5% BSA, sodium azide 0.02%, 0.3% saponin, 1.6% PFA. Labeled samples were acquired on a Helios instrument using a flow rate of 0.030 ml/min. Data were analyzed using FlowJo software (v10.2). At least 500,000 events were acquired for each sample. The CD4 T cell phospho-protein signaling profiles evaluated in ICU and non-ICU individuals were compared to values obtained from healthy individuals (c.f. Study group section).

**Statistical analyses.** Statistical analyses were conducted using R version (v.3.6.3) (The R Foundation for Statistical Computing) and Stata version 16.1 (Stata Corp, College Station, TX, USA). Inter-group clinical data comparisons were performed using chi-square or Fisher's exact test for categorical variables and Kruskal-Wallis test for continuous variables. Descriptive values were presented as violin plots for continuous variables. Serum marker levels and mass cytometry cell population values were log10 transformed for statistical analysis. Statistical significance ($P$ values) was obtained using Kruskal–Wallis test. Bonferroni's correction was applied for multiple comparisons (exact $P$ values are available in Source Data file). Serum markers whose p-value was below the threshold were then considered as candidates for diagnosis of severe (ICU) cases. For each candidate marker, the optimal cutpoint to distinguish between ICU and non-ICU patients was computed using the cutpt command of Stata and default settings (i.e., maximization of the sensitivity × specificity product). Based on the results, the candidate markers were dichotomized into lower than and higher or equal to the cutpoint, and the area under the receiver-operating curve (AUC), the sensitivity, specificity, positive and negative predictive values and the positive likelihood ratio were computed using the roccomp and the diagt commands of Stata. The two markers displaying the best AUCs were then combined into a binary variable (both values high and other) and considered for the classification of the patients. The clinical relevance of the two markers was checked by multivariable analysis using stepwise forward logistic regression using a $p$-value for entry = 0.05 and a p-value for removal = 0.10. Of the initial 207 patients, 197 had vital data at follow-up and 186 had data allowing for survival analysis. Bivariate analysis of the associations between categories of markers and vital status (death/alive) were assessed using chi-square. Survival analysis was conducted using Cox proportional hazards regression, unadjusted or adjusted for age. A second survival analysis was conducted using a multilevel survival model using a Weibull distribution, where patients were nested within each cohort, and adjusting for age (continuous) and ICU stay (yes/no). For the survival analysis, results were expressed as multivariable-adjusted hazards ratio (HR) and 95% CI.

**Reporting summary.** Further information on research design is available in the Nature Research Reporting Summary linked to this article.

## Data availability

The FCS raw data are protected and are not available due to data privacy laws. However, raw data for all Figures, Tables, and Supplementary Figures and Tables are provided with this paper in Source Data file. Source data are provided with this paper.

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

## Acknowledgements

We are grateful to Nathalie Felix, Stéphanie Gregoretti, Aurélie Myard, Nadine Do Rosario, Amélie Muralti, Thibaut Decaillon, Emmanuelle Medjitna, Philippe Kiehl, Gonzalo Tapia, Manuela Lavelli, Michael Moulin, Manon Geiser, Line Esteves-Leuenberger, Olivia Munoz and Riddhima Banga for technical assistance. We are grateful to the members of the French COVID study. The members of the French COVID study group are listed in Supplementary Materials. Rosemary Hottinger and Fabio Candotti for study management. We are grateful to Finally, Aaron Weddle and John Weddle for their assistance with the figures. This work was supported by Swiss National Science Foundation Grants 320030_200912 to M.P. and 31CA30_196852 to G.P.

## Author contributions

M.P., C.F., and G.P. designed the study. M.P., M.S., P.M.-V., and C.F analyzed the data. A.W., Y.L., C.L., J.G., French COVID cohort study group, D.C., T.R., J.R., P.V., G.W., M.O., and T.C. recruited the volunteers, provided the samples and the clinical data. M.S. and P.M.-V. performed the statistical analyses. M.P. and G.P. wrote the manuscript. T.C. provided strategic advice and revised the manuscript. All the authors read and approved the final manuscript.

## Competing interests

The authors declare no competing interests.
