## [Peer Review File · Nature Communications]

REVIEWER COMMENTS

Reviewer #1 (Remarks to the Author):

The prediction of severe disease in patients with COVID-19 is a central clinical challenge, and biomarkers with this ability could help streamline out- and in-hospital patient flow and potentially improve outcome. In addition, very similar challenges are still found in many other infectious diseases, notably bacterial and viral respiratory infections including influenza. The authors present a comprehensive study of immunological parameters and soluble biomarkers in serum with the goal of predicting, upon hospital admission, the need for ICU admission. A particular strength of the study is the use of 2 external validation cohorts (and a large cohort of control subjects), as the result of which the identified HGF and CXCL13 as the most robust and accurate biomarkers. These findings are not completely new, as single center studies have already reported on these markers in COVID-19. However, the combination of the two markers plus the extensive validation constitutes an important advance in our understanding of predictive host biomarkers for severe COVID-19.

Specific comments

Abstract. The text should be restructured in order to flow better. Lines 42-46 are partially redundant, I suggest to move lines 46-48 after the results (line 52). Line 52 tone down to state "HGF may reflect". Line 86, deficient would be more appropriate than immunodeficient

Line 89, do add the respective references

Line 119. Provide more information about these guidelines, as well as the French ones mentioned later on. How do they differ from other internationally used guidelines? Perhaps put the relevant text into the Supplement.

Line 129. Was O2 sat on room air or suppl. O2 (it looks like it was on suppl O2, but please confirm) ?

Line 183 Since the measurements refer to phosphorylated single polypeptides, I don't think the signaling pathways should be named by the phosphorylated protein abbreviations (pNF-kb, pCREB, pERK1/2, pS6 and p38)

Line 238 It is not necessary to repeat numeric values that are already shown in a table.

Fig. 1 I suggest to use a different color for controls so that the contrast to non-ICA is better. Also, could the authors place these (important) individual graphs into the supplement and in Fig. 1 instead show a hierarchical clustering analysis/heatmap plus a selection of the most significant markers? In its current form this figure is too busy and contains too much information that is not necessary for the reader to follow the text.

Fig. 3 It is not necessary to show the ROC curves as a figure since the AUCs are already contained in Table 1. The same applies to the ROC curves shown in the Supplement.

In Discussion the findings should also (albeit briefly) be discussed in the light of the published literature regarding HGF in respiratory infections (pneumonia, influenza)

Frank Pessler

Reviewer #2 (Remarks to the Author):

In this study, Perreau et al. aimed to identify biological signatures that may predict severity of COVID-19 in infected patients proxy by admission to ICU. It was achieved via a case-control design, i.e. by collecting blood samples from patients with PCR-confirmed SARS-CoV-2 infection in the LUH-1 cohort, at admission to ICU (for ICU patients/ severe COVID-19) or to internal medicine ward (for non-ICU patients/ moderate COVID-19), with biomarkers measured by mass cytometry and multiplex beads assay. Levels of each individual biomarker were then dichotomized into either low or high level based on cut-off identified using the Liu method which maximises the product of the sensitivity and specificity. The identified biomarkers, HGF and CXCL13 in combination, were then validated in two other cohorts (FCS and LUH-2), for the prediction to admission to ICU. Separately, they also showed that HGF and CXCL13 in combination may predict 30-day mortality among COVID-19 patients.

Other studies have evaluated the use of peripheral blood biomarkers that may be used to predict subsequent COVID-19 severity, which are commonly proxied by clinical outcomes such as acute

respiratory distress syndrome (ARDS) or death (Del Valle et al Nat Med 2020; Ghazavi et al Cytokine 2021; Balfanz et al PLoS One 2021), need for intensive care such as intubation, mechanical ventilation and ECMO (Ghazavi et al Cytokine 2021; Dorgham J Allergy Clin Immunol 2021), and clinical and laboratory markers of disease severity (Del Valle et al Nat Med 2020). In this study, it was unclear from the manuscript how severity was defined, i.e. what were the criteria of admission to ICU, as admission to ICU was only described as following the recommendation of the local health authorities without providing further details. Furthermore, the use of mass cytometry in measuring biomarkers makes it less transferrable to actual clinical settings.

Other questions/ suggestions:

- line 89: typo
- line 102: for the individuals included in the non-ICU group, i.e. admission to the internal medicine ward (initially), could you please confirm that they had not been admitted to ICU subsequently?
- line 119, 224: please elaborate on the criteria for admission to the ICU as recommended by the guidelines of the public health authorities
- line 356: could you please confirm whether the exclusion criteria for healthy individuals (e.g. sign of hepatitis) applied not only in the healthy individuals group but also in the ICU and non-ICU groups (to make the groups more comparable)?
- line 413: please consider to move this whole section on antibody panels (which spanned 3 pages) to supplementary materials and keep only the essential information

Reviewer #1 (Remarks to the Author):

The prediction of severe disease in patients with COVID-19 is a central clinical challenge, and biomarkers with this ability could help streamline out- and in-hospital patient flow and potentially improve outcome. In addition, very similar challenges are still found in many other infectious diseases, notably bacterial and viral respiratory infections including influenza. The authors present a comprehensive study of immunological parameters and soluble biomarkers in serum with the goal of predicting, upon hospital admission, the need for ICU admission. A particular strength of the study is the use of 2 external validation cohorts (and a large cohort of control subjects), as the result of which the identified HGF and CXCL13 as the most robust and accurate biomarkers. These findings are not completely new, as single center studies have already reported on these markers in COVID-19. However, the combination of the two markers plus the extensive validation constitutes an important advance in our understanding of predictive host biomarkers for severe COVID-19.

Specific comments

1. Abstract. The text should be restructured in order to flow better. Lines 42-46 are partially redundant, I suggest to move lines 46-48 after the results (line 52). Line 52 tone down to state "HGF may reflect".
According to Reviewer's request, the Abstract was restructured in the revised version of the manuscript.
2. Line 86, deficient would be more appropriate than immunodeficient
According to Reviewer' suggestion, the sentence was modified in the revised version of the manuscript.
3. Line 89, do add the respective references
The references were added in the revised version of the manuscript.
4. Line 119. Provide more information about these guidelines, as well as the French ones mentioned later on. How do they differ from other internationally used guidelines? Perhaps put the relevant text into the Supplement.
According to Reviewer's request, the criteria for severe COVID-19 admission in intensive care unit (ICU) or internal medicine ward (non-ICU) were added in the "Methods" section of the revised version of the manuscript.
5. Line 129. Was O2 sat on room air or suppl. O2 (it looks like it was on suppl O2, but please confirm) ?
The reviewer is right, the values correspond to SpO2 during the highest oxygen provision. The text in Supplemental Table 1 has been modified accordingly.
6. Line 183 Since the measurements refer to phosphorylated single polypeptides, I don't think the signaling pathways should be named by the phosphorylated protein abbreviations (pNF- κ b, pCREB, pERK1/2, pS6 and p38)
According to Reviewer' suggestion, the sentence was modified in the revised version of the manuscript.
7. Line 238 It is not necessary to repeat numeric values that are already shown in a table.
According to Reviewer' suggestion, the numeric values were removed from the revised version of the manuscript.
8. Fig. 1 I suggest to use a different color for controls so that the contrast to non-ICA is better. Also, could the authors place these (important) individual graphs into the supplement and in Fig. 1 instead show a hierarchical clustering analysis/heatmap plus a selection of the most significant markers? In its current form this figure is too busy and contains too much information that is not necessary for the reader to follow the text.

According to Reviewer's request, the colors corresponding to healthy individuals, non-ICU and ICU patients contrast better in the revised version of the manuscript. In addition, a hierarchical clustering analysis/heatmap has been created (Figure 2A). Finally, only a selection of the most significant markers (N=18) is depicted in Figure 2B. The remaining markers (N=31) are depicted in Supplemental Figure 3 in the revised version of the manuscript.

9. Fig. 3 It is not necessary to show the ROC curves as a figure since the AUCs are already contained in Table 1. The same applies to the ROC curves shown in the Supplement. According to Reviewer's suggestion, Figure 3, Supplemental Figure 4 and corresponding Figure legends were removed from the revised version of the manuscript.
10. In Discussion the findings should also (albeit briefly) be discussed in the light of the published literature regarding HGF in respiratory infections (pneumonia, influenza) According to Reviewer's request, the role of HGF in respiratory infections (pneumonia, influenza) is now briefly discussed in the discussion section of the manuscript.

Reviewer #2 (Remarks to the Author):

In this study, Perreau et al. aimed to identify biological signatures that may predict severity of COVID-19 in infected patients proxy by admission to ICU. It was achieved via a case-control design, i.e. by collecting blood samples from patients with PCR-confirmed SARS-CoV-2 infection in the LUH-1 cohort, at admission to ICU (for ICU patients/ severe COVID-19) or to internal medicine ward (for non-ICU patients/ moderate COVID-19), with biomarkers measured by mass cytometry and multiplex beads assay. Levels of each individual biomarker were then dichotomized into either low or high level based on cut-off identified using the Liu method which maximises the product of the sensitivity and specificity. The identified biomarkers, HGF and CXCL13 in combination, were then validated in two other cohorts (FCS and LUH-2), for the prediction to admission to ICU. Separately, they also showed that HGF and CXCL13 in combination may predict 30-day mortality among COVID-19 patients.

Other studies have evaluated the use of peripheral blood biomarkers that may be used to predict subsequent COVID-19 severity, which are commonly proxied by clinical outcomes such as acute respiratory distress syndrome (ARDS) or death (Del Valle et al Nat Med 2020; Ghazavi et al Cytokine 2021; Balfanz et al PLoS One 2021), need for intensive care such as intubation, mechanical ventilation and ECMO (Ghazavi et al Cytokine 2021; Dorgham J Allergy Clin Immunol 2021), and clinical and laboratory markers of disease severity (Del Valle et al Nat Med 2020).

1. In this study, it was unclear from the manuscript how severity was defined, i.e. what were the criteria of admission to ICU, as admission to ICU was only described as following the recommendation of the local health authorities without providing further details.

We are grateful to the reviewer for rising this major point. This study focused on individuals suffering from severe COVID-19 that required hospitalization. The major objective of this study was to identify biological signatures of severe COVID-19 predictive of admission in intensive care unit. Therefore, we compared individuals with severe COVID who were admitted in ICU or not. The criteria were the following: individuals with severe COVID-19 with acute respiratory failure requiring mechanical ventilation and/or cardio-circulatory insufficiency requiring the administration of

vasoactive agents were admitted to ICU. Patients admitted to the internal medicine ward (non-ICU) required supplemental oxygen and did not had criteria for ICU admission. As requested by the reviewer, the criteria for severe COVID-19 admission in intensive care unit (ICU) or internal medicine ward (non-ICU) were implemented in the method section of the revised version of the manuscript.

With regard the guidelines of the Swiss Society of Internal Medicine and of Swiss Society of Intensive Care, the criteria limiting the admission to the ICU included:

- a) Willingness of the patient
- b) Refractory shock to noradrenaline
- c) Metastatic cancer with life expectation <1 year
- d) Terminal neurodegenerative disease with sever dementia
- e) Severe and irreversible CNS disease
- f) Hearth failure NYHA IV
- g) Child-Pugh Cirrhosis >8 (known)

2. Furthermore, the use of mass cytometry in measuring biomarkers makes it less transferrable to actual clinical settings.

We fully agree with the reviewer's comment. The aim of the present study was to first explore a large number (over 170) cellular and serum markers. Therefore, the use of a mass cytometry-based strategy was chosen because it allowed investigating more than 40 parameters simultaneously. On the event of an identification of cellular marker(s) and/or specific cell population(s) significantly associated with ICU admission, a flow cytometry based approached could have been proposed with a more narrowed antibody panel.

3. line 89: typo

We thank the reviewer for rising this typo error. The references were added in the revised version of the manuscript.

4. line 102: for the individuals included in the non-ICU group, i.e. admission to the internal medicine ward (initially), could you please confirm that they had not been admitted to ICU subsequently?

1/91 non-ICU patients enrolled in the LUH-1 and LH-2 cohorts was admitted to the ICU three days after hospitalization. 4/31 non-ICU patients enrolled in the French cohort were admitted to the ICU. 1 patient after one day, 1 after 3 day, 1 after 6 days and 1 after 7 days.

All these patients did not have the clinical criteria for ICU admission at the day of hospitalization. We believe that these observations on the this handful number of patients further enhance the potency of HGF and CXCL13 as predictor factors distinguishing between non-ICU versus ICU patients.

5. line 119, 224: please elaborate on the criteria for admission to the ICU as recommended by the guidelines of the public health authorities

As mentioned in Reviewer #2 point #1, the criteria were the following: individuals with severe COVID-19 with acute respiratory failure requiring mechanical ventilation and/or cardio-circulatory insufficiency requiring the administration of vasoactive agents were admitted to ICU. Patients admitted to the internal medicine ward (non-ICU) required supplemental oxygen and did not had criteria for ICU admission. As requested by the reviewer, the criteria for severe COVID-19 admission in intensive care unit (ICU) or internal medicine ward (non-ICU) were implemented in the method section of the revised version of the manuscript.

6. line 356: could you please confirm whether the exclusion criteria for healthy individuals (e.g. sign of hepatitis) applied not only in the healthy individuals group but also in the ICU and non-ICU groups (to make the groups more comparable)?

We did not exclude SARS-CoV-2 infected individuals admitted or not in ICU based on the exclusion criteria used for the selection of the “healthy group”. Indeed, the main objective of this study was to identify biological signatures of severe COVID-19 predictive of admission in intensive care unit. We therefore first defined the reference values of each parameter tested using the “healthy group”. These reference values had to be defined on individuals with no sign of acute or chronic infection, prior diagnosis of autoimmune disease or primary or secondary immunodeficiency and current or past use of medications that are known modify the immune response. Pregnancy was the only exclusion criteria for SARS-CoV-2 infected individuals enrolled in the present study. We however agree with reviewer’s comment. Such an approach might have increased the variability within the “non-ICU” and “ICU” groups. On the other hand, one could also argue that this approach reinforce the signatures that were identified despite this potential variability.

7. line 413: please consider to move this whole section on antibody panels (which spanned 3 pages) to supplementary materials and keep only the essential information

The antibody panel section of the methods has been moved to the supplementary material in the revised version of the manuscript.

REVIEWER COMMENTS

Reviewer #1 (Remarks to the Author):

The authors have addressed my comments to my satisfaction. Just one detail: I did not mean to delete the information in lines 46-48, but I suggest to place it at the end of the current line 52. ADM should be spelled out on first mention in the abstract and in the title.

Reviewer #2 (Remarks to the Author):

Thank you for addressing my questions.

For your response #6 regarding the difference in exclusion criteria for healthy/non-ICU/ICU groups, please consider adding the part about using the healthy group to define reference value in the Methods section (at which step is it done - mass cytometry or statistical analyses?), and its implications to your study results and interpretation in the Discussion section.

REVIEWERS' COMMENTS

Reviewer #1 (Remarks to the Author):

The authors have addressed my comments to my satisfaction. Just one detail: I did not mean to delete the information in lines 46-48, but I suggest to place it at the end of the current line 52. ADM should be spelled out on first mention in the abstract and in the title.

According to Reviewer's request, the Abstract was restructured in the revised version of the manuscript.

Reviewer #2 (Remarks to the Author):

Thank you for addressing my questions.

For your response #6 regarding the difference in exclusion criteria for healthy/non-ICU/ICU groups, please consider adding the part about using the healthy group to define reference value in the Methods section (at which step is it done - mass cytometry or statistical analyses?), and its implications to your study results and interpretation in the Discussion section.

As indicated in the manuscript, the upper normal values for each serum concentration of cytokines, soluble cytokine receptors, chemokines and growth factors were defined based on the results obtained in the 450 sera collected from healthy individuals (mean + 2 standard deviations) (lines 376-378). In addition, according to Reviewer's request, the method and discussion sections were modified accordingly.